# Pulmonary Lesion Classification Framework Using the Weighted Ensemble Classification with Random Forest and CNN Models for EBUS Images

**DOI:** 10.3390/diagnostics12071552

**Published:** 2022-06-26

**Authors:** Banphatree Khomkham, Rajalida Lipikorn

**Affiliations:** Machine Intelligence and Multimedia Information Technology Laboratory (MIMIT), Department of Mathematics and Computer Science, Faculty of Science, Chulalongkorn University, Bangkok 10330, Thailand; banphatree.kh@gmail.com

**Keywords:** pulmonary lesion, endobronchial ultrasonography images (EBUS), convolutional neural network (CNN), radiomics features, random forest, gray-level co-occurrence matrix (GLCM), weighted ensemble

## Abstract

Lung cancer is a deadly disease with a high mortality rate. Endobronchial ultrasonography (EBUS) is one of the methods for detecting pulmonary lesions. Computer-aided diagnosis of pulmonary lesions from images can help radiologists to classify lesions; however, most of the existing methods need a large volume of data to give good results. Thus, this paper proposes a novel pulmonary lesion classification framework for EBUS images that works well with small datasets. The proposed framework integrates the statistical results from three classification models using the weighted ensemble classification. The three classification models include the radiomics feature and patient data-based model, the single-image-based model, and the multi-patch-based model. The radiomics features are combined with the patient data to be used as input data for the random forest, whereas the EBUS images are used as input data to the other two CNN models. The performance of the proposed framework was evaluated on a set of 200 EBUS images consisting of 124 malignant lesions and 76 benign lesions. The experimental results show that the accuracy, sensitivity, specificity, positive predictive value, negative predictive value, and area under the curve are 95.00%, 100%, 86.67%, 92.59%, 100%, and 93.33%, respectively. This framework can significantly improve the pulmonary lesion classification.

## 1. Introduction

Cancer has been regarded as the leading cause of death among the world’s population from past to present, and its prevalence is expected to rise steadily. Among many types of common cancers, lung cancer is the leading cause of death, followed by colorectal, liver, stomach, and female breast cancers. According to the International Agency for Research on Cancer, there were 2.2 million new cases of lung cancer diagnosed and 1.8 million deaths globally in 2020 [1]. The majority of lung cancer patients do not show symptoms until the disease has advanced, but some early lung cancer patients may show the symptoms; therefore, early diagnosis can lower the mortality rate significantly [2]. Furthermore, lung cancer can be cured, and treatment is more effective if it is detected early [3,4]. In general, there are many techniques for diagnosis and staging of lung cancer such as computed tomography (CT), positron emission tomography—computed tomography (PET–CT), magnetic resonance imaging (MRI), and EBUS [5,6,7]. EBUS has become popular in recent years since this technique utilizes no radiation and scans in real time. It is the most recent screening technology for obtaining small wounds with minimal pain [8]. Although EBUS is a good way to detect lung cancer early, its performance is limited by tissue superposition, which can result in false-negative diagnoses [9].

In clinical research, many researchers attempt to find criteria to distinguish pulmonary lesions in EBUS images by using both retrospective and prospective methods [10,11,12,13]. According to previous research [14], the characteristics of malignant lesions in EBUS images have a heterogeneous pattern, a short axis, presence of coagulation necrosis sign, round shape, distinct margin, and absence of central hilar structure, while the characteristics of benign lesions in EBUS images show the presence of calcification, nodal conglomeration, and echo intensity. As a result, in visual tasks, precise and reliable EBUS interpretation and lung cancer diagnosis are extremely challenging and also depend on the skills and experiences of radiologists. Therefore, several computer-aided diagnosis (CAD) methods have been proposed to address this problem.

Morikawa et al. [15] studied 30 malignant and 22 benign EBUS images from 60 patients who were subjected to a bronchoscopy using histogram-based quantitative evaluation of EBUS images. The regions of interest (ROIs) inside EBUS images were suitably selected by experimenting with a phantom model submerged in water to extract six histogram features. The extracted features of EBUS images were distinguished by using Mann–Whitney U tests.

Alici et al. [16] processed 1051 lymph nodes from 532 patients by using the sonographic features such as grayscale, echogenicity, shape, size, margin, presence of necrosis, presence of calcification, and absence of central hilar structure via EBUS images. Decision tree analysis was applied to discriminate lymph nodes between benign and malignant.

Khomkham and Lipikorn [17] proposed two robust features that were extended from a gray-level co-occurrence matrix (GLCM) as well as a technique for lung cancer classification utilizing a genetic algorithm and support vector machines (SVM). The classification performance with accuracy, sensitivity, specificity, and precision is 86.52%, 87.27%, 85.29%, and 90.57%, respectively.

Gómez et al. [18] studied the performance of 22 co-occurrence statistics in conjunction with six gray-scale quantization levels to identify breast lesions on ultrasound (BUS) images. The 436 BUS images were utilized in this study; the number of carcinoma lesions was 217 and the number of benign lesions was 219. The best area under the curve obtained from using 32 gray levels and 109 features was 0.81.

Radiomics analysis is also widely used in cancer diagnosis [15,16,17,18]. The concept of radiomics analysis is to extract a massive number of quantitative features from medical images by using shape features, first order features, second order features, or higher order features. In recent years, deep learning (DL) methods have been used tremendously in computer vision aided by advances in computation and very large amounts of data. In comparison to traditional machine learning, deep learning can accurately detect appropriate features for particular classification tasks and possibly clarify feature selection problems without the need for complicated image processing pipelines and pattern recognition procedures. As a superb method in DL technology, convolutional neural network (CNN) has been significantly improved in image classification and object detection, including medical imaging and it is now one of the dominant methods. CNN has been applied to medical images to solve many different problems.

Jia et al. [19] presented a novel framework for properly classifying cervical cells based on the strong feature CNN–support vector machine (SVM) model. The technique was developed for merging the strong features recovered by GLCM and Gabor with abstract features acquired from CNN’s hidden layers. Their method outperformed state-of-the-art models with 99.3 percent of accuracy.

Tan et al. [20] proposed a modified CNN-based 3D-GLCM to classify polyps in colonography. This model could handle a small number of datasets by using the advantage of GLCM features. The experimental results show that CNN learning from GLCMs outperforms CNN on raw CT images in terms of classification performance. The model achieves up to 91 percent accuracy by using two-fold cross-validation.

Islam et al. [21] created a deep learning approach consisting of the combination of CNN and long short-term memory (LSTM) to autonomously diagnose COVID-19 via X-ray images. The CNN was utilized for deep feature extraction, while LSTM was used for standard feature extraction and COVID-19 diagnosis. The experimental results reveal that the suggested method obtained an accuracy of 99.4 percent.

Li et al. [22] used chest X-ray (CXR) images to assess the predictive performance of DL models in the recognition and classification of pneumonia. In the pooling step, they utilized bivariate linear mixed models. The results demonstrate that DL performed well in differentiating bacterial from viral pneumonia and in categorizing pneumonia from normal CXR radiographs.

Zhang et al. [23] developed a ResNet model for medical picture classification in smart medicine by replacing global average pooling with adaptive dropout. The results of the experiments on a GPU cluster indicate that the provided model delivered excellent recognition performance without a significant loss in efficiency.

Cai et al. [24] developed a mask region–convolutional neural network (Mask R–CNN) and ray-casting volume rendering algorithm-based detection and segmentation techniques for lung nodule 3D visualization diagnosis. Mask R–CNN of weighted loss achieved sensitivities of 88.1 percent and 88.7 percent, respectively.

Wang et al. [25] presented a new multiscale rotation-invariant convolutional neural network (MRCNN) model for identifying different kinds of lung tissue using high-resolution computed tomography. The suggested technique outperformed the most recent findings on a public interstitial lung disease database.

Anthimopoulos et al. [26] proposed to use a deep CNN to categorize patch-based CT image into seven groups, containing six distinct interstitial lung disease patterns and healthy tissue. A new network architecture was created to capture the low-level textural characteristics of lung tissue. According to the experiments, the categorization performance was around 85.5 percent.

In 2019, Chen et al. [27] proposed the CAD system for differentiating lung lesions via EBUS images using CNN. Because the dataset was small, data augmentation was performed by flipping and rotating images. Then the fine-tuned CaffeNet–SVM was used to differentiate lung lesions. The experimental results revealed that the proposed system to achieve up to 85.4 percent accuracy.

In 2021, Lei et al. [28] proposed a low-dose CT image denoising method for improved performance of lung nodule classification. Because scans have substantial noise, they have significant influence on lung nodule classification. The proposed method enables cooperative training of image denoising and lung nodule classification by utilizing self-supervised loss and cross-entropy loss. According to the experiments, the simultaneous training of image denoising and lung nodule classification increases the performance significantly.

Lei et al. [29] proposed a novel method for analyzing shape nodule with a CNN using soft activation mapping. Soft activation mapping captures more fine-grained and discrete attention regions to locate the low-grade malignant nodule. The results of the experiments on the LIDC–IDRI dataset revealed that the proposed method outperformed state-of-the-art models in terms of false positive rate.

Ensemble methods are techniques for developing multiple models and then combining them to produce better results. Moreover, when compared to a single model, ensemble approaches often produce more accurate results. Recently, an ensemble method has been reported in a variety of fields. The ensemble method has been applied to medical images to solve many different problems. Guo et al. [30] proposed an ensemble learning method for COVID-19 diagnosis via CT obtained by using ordinal regression. This model could enhance classification accuracy by learning both intraclass and interclass links between phases. The experimental results revealed that as modified ResNet-18 was utilized as the backbone; accuracy rose by 22% when compared to standard approaches.

However, most of the existing techniques need large datasets to yield satisfactory results. Thus, this paper proposes a novel pulmonary lesion classification framework that does not need a large training dataset by combining radiomics features and patient data with standard features that are extracted from EBUS images as input data, then using random forest, CNN, and weighted ensemble to classify pulmonary lesions.

The structure of this paper is as follows: Section 2 describes the details of the materials; Section 3 explains the proposed framework; the results and discussion are summarized in Section 4; and Section 5 provides the conclusion.

## 2. Materials

The data used for evaluation of the proposed framework consist of both EBUS images and patient data. The EBUS images were obtained by skilled radiologists from Phramongkutklao Hospital, Bangkok, Thailand, between November 2011 and May 2016. The EBUS images were obtained using an endoscopic ultrasound system (EU-ME1; Olympus) and a 20 MHz miniature radial probe (UM-S20-17S; Olympus). The probe provides a 360-degree panoramic ultrasonic view of the lesion. The radiologists collected 200 EBUS images from 200 patients. There are 124 malignant lesions and 76 benign lesions. The image format is an 8-bit RGB image. The size of each image was cropped to 776 × 776 pixels. Examples of different pulmonary lesion patterns in EBUS images are shown in Figure 1.

For patient data, both numerical and categorical data that were used consist of gender (male, female), smoking history (smoker, no smoker, and ex-smoker), age (between 17 and 86), and lesion size (less than 3 cm, more than or equal to 3 cm) as shown in Table 1.

## 3. Methods

The proposed pulmonary lesion classification framework is shown in Figure 2. The framework is based on the integration of three modified machine learning models and the weighted ensemble classification. The three modified machine learning models are the radiomics feature and patient data-based model, the single image-based model, and the multi-patch-based model.

### 3.1. Preprocessing

The preprocessing step consists of class balancing, mask generation, feature extraction, and window of interest (WOI) selection. The class balancing was performed to generate more data and balance the amount of training data for the models since our dataset contains only 200 EBUS images. Then the EBUS images were converted to grayscale images to be used in mask generation and the radiomics feature and patient data-based model, while the original RGB EBUS images were used in the single image-based model and the multi-patch-based model. Mask generation was performed to define the region of lesion for the single image-based model and the multi-patch-based model. The window of interest selection was performed to divide the region of lesion into small patches which were used for the multi-patch-based model.

#### 3.1.1. Class Balancing

Because the dataset is too small and there are more malignant images than benign images, image augmentation, which is an effective way to increase the amount of data without having to obtain new images, was performed to balance the data. The augmentation methods used in this paper are rotation, vertical flipping, and horizontal flipping to preserve the main characteristics of the images. The images were rotated by 90 and 180 degrees, and they were also flipped vertically and horizontally, as shown in Figure 3.

Data augmentation combines both malignant strategy and benign strategy [26] to balance data in both training classes; therefore, the images in malignant class were rotated but were not flipped because there were more malignant images than benign images. From 200 EBUS images, 160 images (80% of the dataset) were used as training data and 40 images (20% of the dataset) were used as test data. After augmentation, the total number of training data for both classes increased from 160 to 602. The number of augmented malignant and benign images is shown in Table 2.

#### 3.1.2. Mask Generation

In this step, the mask that was used to represent the region of lesion within the boundary in each EBUS image was generated. The mask generation consists of two main parts: image enhancement and boundary detection. In the medical field, many techniques have been introduced to enhance image quality [31,32]. Contrast stretching (CS) [33] is one of the enhancement techniques that is used to deal with adjusting contrast and improving image quality in the region of interest. By using CS, the bright components can be made brighter, while the dark background can be made darker. CS operation on an image is shown in Equation (1):(1)$$\bar{I}\left(x,y\right)=\{\begin{array}{cc} 0, & I\left(x,y\right)<L\\ I{\left(x,y\right)}^{\gamma }, & L\leq I\left(x,y\right)\leq H\\ 1, & I\left(x,y\right)>H \end{array}$$
where I(x,y) is the original image, (x, y) are the coordinates of a pixel, $\bar{I}\left(x,y\right)$ is the enhanced image, L is the low threshold intensity, H is the high threshold intensity, and γ is a constant value.

After enhancing the images, the next step is boundary detection. There are many boundary detection techniques [34,35,36,37] that can be applied, and the technique called ray tracing is the technique that was used to detect the lesion boundary in this paper [37]. Once the boundary was detected, the mask of the original image was generated by assigning 1 to the area inside the boundary and 0 to the area outside the boundary. Figure 4b shows the mask of the original image in Figure 4a. Figure 4c shows the region of lesion that can be obtained by performing AND operation between Figure 4a,b.

#### 3.1.3. Feature Extraction

There are several other important features that can be extracted from EBUS images and can be utilized for lesion classification and texture analysis. Feature extraction is performed to improve the performance of the classifier by searching for the most condensed and informative set of features. Radiomics features are widely used in many fields of pattern recognition, computer vision, and image classification. In this paper, the radiomics features were extracted from the area of lesion inside the boundary, as shown in Figure 4c. The radiomics features, which consist of six classes: shape-based 2D (9 features), GLCM (24 features), gray-level dependence matrix (GLDM) (14 features), gray-level run length matrix (GLRLM) (16 features), gray level size zone matrix (GLSZM) (16 features), and neighboring gray tone difference matrix (NGTDM) (5 features), were extracted using the pyradiomics package [38]. Another GLCM feature known as the adaptive weighted-sum of the upper and lower triangular GLCM or AWS is also included in the radiomics features [17]. This feature is effective at determining heterogeneity, which is one of the most important characteristics of malignancy. Besides radiomics features, four features were extracted from patient data: gender, smoking history, age, and lesion size. The total number of features is 89 features. All features used in this paper and their correlations are shown using the correlation heat map [39] in Figure 5.

#### 3.1.4. WOI Selection

The last step of preprocessing is WOI selection which prepares the input data for the multi-patch-based model. The WOI selection divides a lesion into small patches or windows. The study by Morikawa et al. [15] found that the most suitable region of lesion is the ring between 2 mm to 5 mm from the probe, thus the patches were selected from this ring. The size of the patch was derived from the size of the biggest square window that can fit within the ring which is 32 × 32 pixels. The patches in each ring are all the windows that can be tiled inside the ring area as shown in Figure 4d.

### 3.2. The Proposed Framework

The proposed framework consists of three machine learning models that were used to calculate the probability of being benign or malignant. The first model is based on radiomics features and patient data, the second model is based on the original EBUS images, and the third model is based on multiple patches of lesion.

#### 3.2.1. Radiomics Feature and Patient Data-Based Model

The first model consists of feature selection and classification as shown in Figure 6. Feature selection was performed to reduce the number of features that are redundant and irrelevant. Mutual information (MI) criterion [40] which is one of the feature selection techniques was used to select relevant features from radiomics features and patient data. MI between feature and target class is a non-negative value that measures dependency. It is equal to zero if and only if two variables are independent; higher value means higher dependency.

A subset of selected features that were obtained after applying mutual information criterion were used as input to the random forest classifier (RF) [41]. RF is a supervised machine learning classifier that is composed of as many decision trees on different samples as possible and combines the output from all the trees. RF can decrease overfitting problems in decision trees, as well as variation, and hence improve accuracy. RF is also one of a few classifiers that can handle both categorical and numerical features. RF is trained on a subset of selected features that contains both patient data and radiomics features. The output is the probability of being benign or malignant, $P_{1}$.

#### 3.2.2. Single Image-Based Model

The second model uses the original EBUS images as input data for the fine-tuned dense convolutional network 169 (DenseNet) [42]. DenseNet feature extractor was used to extract both local and global characteristics from an image. These local characteristics focus on the patterns of texture; i.e., homogeneity, heterogeneity, hyperechoic dot, hyperechoic arc, anechoic area, and linear air bronchogram while global characteristics focus on shape, size, and patterns of the texture of the whole lesion. DenseNet 169 architecture connects all layers densely. Each layer receives input from the preceding layers and forwards its output to the subsequent layers via its feature map. Its goal is to remove the redundant layer. Each layer inherits collective knowledge from the layers before it. Consequently, the classification layer receives data from all of the preceding layers as input data. DenseNet169 can produce excellent results, but fine-tuning their hyper parameters requires expert knowledge, a large dataset, and a significant amount of time, thus transfer learning [43] is used to solve such problems. DenseNet 169 can reuse the previously trained model. The idea behind transfer learning is to use a complicated and effectively pre-trained model, such as ImageNet, and then apply the learned knowledge to a new problem with a small dataset (EBUS images for this paper). DenseNet 169 is trained from ImageNet [44] and the weights from the first convolutional layer in block 1 to the last convolutional layer in block 8 are frozen. The classification layer was trained by EBUS images, separately. The output layer of the fine-tuned DenseNet 169 for the single image-based model returned the probability of being benign or malignant, $P_{2}$. The architecture of the fine-tuned DenseNet 169 for the single image-based model is shown in Figure 7 and the hyper-parameters of the model are shown in Table 3.

#### 3.2.3. Multi-Patch-Based Model

The third model is called the multi-patch-based model because it uses all patches of size 32 × 32 pixels from each image as input to the proposed CNN. Since the input of this model was the patch, the CNN feature extractor was used to extract only local characteristics. The proposed CNN architecture for multi-patch images is shown in Figure 8.

The architecture of the proposed CNN is shown in Figure 9. The input is convolved by a series of four convolutional layers. The size of kernels of these convolutional layers is set to 3×3. The numbers of kernels of four convolutional layers are 8, 16, 32, and 64, respectively, as shown in Table 4. Every convolutional layer is followed by ReLU activation and Max pooling. The kernel size of Max pooling layers is set to 2×2 with no padding.

The batch size which defines the number of samples that are propagated through the network is set to 128. Dropout and batch normalization are also applied to prevent overfitting problems. The two-dimensional output is flattened and SoftMax activation is used to calculate the categorical probability distribution. The hyper-parameters of CNN architecture are shown in Table 5.

Figure 10 depicts how we visualize the learned features. Although there are no discernible structures, they are useful for classifying the texture of pulmonary lesions. Since each image contains the classification results of multiple patches, the classification result for each image can be obtained by using the decision threshold. The decision threshold is used to classify whether a lesion in an image is benign or malignant by calculating the probability of being malignant from the ratio of the patches that are classified as malignant to the total number of patches of an image as defined by Equation (2).
(2)$$P_{3}\left(I\right)=\frac{n_{M}}{n_{B}+n_{M}}$$
where $n_{M}$ is the number of malignant patches, $n_{B}$ is the number of benign patches. If the probability is less than the decision threshold value, T, then a lesion is classified as benign; otherwise, malignant as defined by Equation (3).
(3)$$Class\left(I\right)=\{\begin{array}{cc} \;1 & if\;P_{3}\left(I\right)>T\\ \;0 & otherwise \end{array}$$
where $P_{3}\left(I\right)$ is the probability of being malignant I. Class 0 represents benign, and class 1 represents malignant.

### 3.3. Weighted Ensemble Classification

The last step of the framework is to finally classify a lesion using the weighted ensemble classification [45] with the probability distributions from the three models as defined by Equation (4):(4)$$P\left(I\right)=w_{1}P_{1}\left(I\right)+w_{2}P_{2}\left(I\right)+w_{3}P_{3}\left(I\right),$$
(5)$$w_{1}+w_{2}+w_{3}=1$$
where P(I) is the probability of being malignant, $w_{1},\;w_{2}$, and $w_{3}$ are the weight of each model with the sum of these three weights equal to 1. $P_{1}$ is the probability from the radiomics feature and patient data-based RF, $P_{2}$ is the probability from the single image-based CNN, and $P_{3}$ is the probability from the multi-patch-based CNN. If P(I) is less than the cutoff value then a lesion is benign; otherwise, malignant. The optimal cutoff value is defined by the value that yields the highest accuracy during the training.

### 3.4. Performance Evaluation

The proposed pulmonary lesion classification framework is evaluated on the dataset that is randomly partitioned into two sets of 80:20. The training set consists of 80% of the data, while the test set consists of the remaining 20% of the data.

The performance is measured using six statistical indicators: accuracy (Acc) sensitivity (Sen), specificity (Spec), positive predictive value (PPV), negative predictive value (NPV), and area under the curve (AUC) as defined by Equations (6)–(10).
(6)$$Acc=\frac{correctly\;detected\;cases}{total\;cases\;\;}$$
(7)$$Sen=\frac{correctly\;detected\;malignant\;cases}{total\;malignant\;cases}$$
(8)$$Spec=\frac{correctly\;detected\;benign\;cases}{total\;benign\;cases}$$
(9)$$PPV=\frac{correctly\;detected\;malignant\;cases}{detected\;malignant\;cases}$$
(10)$$NPV=\frac{correctly\;detected\;benign\;cases}{detected\;benign\;cases}$$

## 4. Experimental Results and Discussion

This section presents the experimental setup and the experimental results with discussion.

### 4.1. Experimental Setup

All the experiments were performed on a workstation (Intel (R) Core (TM) 3.00 GHz processor with 16 GB of RAM) and a NVIDIA GeForce GTX1650GPU. For preprocessing, the experiments were performed using MATLAB R2020b. For the classification, the experiments were implemented by python programming language with python libraries such as Keras, pandas, Scikit-learn, and NumPy.

### 4.2. Experimental Results

The results of EBUS image enhancement, feature selection for the radiomics feature and patient data-based model, and the classification results of the proposed framework are described in this section.

#### 4.2.1. EBUS Image Enhancement

To improve the quality of all EBUS images, the parameter setting for CS includes  L and H which were determined by sorting the intensity values of an image. From our dataset, the optimal values for L and H were at 1 percentile and 99 percentiles of intensity values. The enhanced images and their histograms are shown in Figure 11. After EBUS image enhancement was performed, more details of lesion components can be clearly seen. Figure 11a depicts the original image, while Figure 11c depicts the enhanced image. The histograms of these two images show that the range of intensity values was widened after using CS.

#### 4.2.2. Feature Selection

Mutual information was performed on radiomics features and patient data to select only relevant features that are necessary for radiomics features and patient data-based model, and the most effective number of selected features was 57 out of 89 features. Figure 12 shows 57 features that were selected from both radiomics features and patient data with MI scores greater than zero.

#### 4.2.3. Classification Performance

The performance of each model and the proposed classification framework were evaluated. For the radiomics feature and patient data-based model, the most suitable learning parameters were determined through the training using data from 200 patients who have both EBUS images and patient data. The RF classifier was performed on 57 features that were selected by MI. The forest’s tree number was set to 1000, the Gini index was used as the split quality measure, and the minimum number of samples required to divide an internal node was set to two. Table 6 displays the RF performance. It can be seen that the RF performance can achieve up to 85% of accuracy. The subset of relevant features consisting of three patient data and 54 out of 85 radiomics features were chosen. This indicates that radiomics features and patient data are important in the analysis of pulmonary lesions.

Figure 13a shows the confusion matrix of radiomics feature and patient data-based model. From the test set of 40 EBUS images, two lesions out of 25 malignant lesions were misclassified as benign, while four lesions out of 15 benign lesions were misclassified as malignant.

Figure 14 depicts the misclassification results from the radiomics feature and patient data-based model. Figure 14a shows a malignant lesion that was misclassified as benign because the texture of the lesion is homogeneous with no echoic arc and echoic dot, whereas a benign lesion in Figure 14b was misclassified as malignant because its texture is heterogeneous, which is a common characteristic of malignant lesions, thus making it difficult to classify correctly.

For the single image-based model, the original EBUS images of 200 patients were augmented to obtain 602 images (305 images in the benign class and 297 images in the malignant class). Figure 13b depicts the confusion matrix of the single image-based model where three malignant lesions were misclassified as benign, and seven benign lesions were misclassified as malignant.

Figure 15 depicts the misclassification of the single image-based model. Figure 15a shows a malignant lesion that was misclassified as benign because the texture of the lesion was quite smooth, which is a common feature of benign lesion. Figure 15b shows a benign lesion that was misclassified as malignant because its texture is heterogeneous.

Next, the multi-patch-based model used the same image data of 200 patients that were used for the single image-based model, but each image was divided into patches. Depending on the region of lesion, the number of patches in each EBUS image can be from one to fourteen. After augmentation and WOI selection, 6795 patches were obtained with 3335 benign patches and 3460 malignant patches. After the model was trained by 6369 patches, each patch extracted from the test image was classified independently. The classification result of each image was determined from the results of all patches using the decision threshold. In this experiment, T was set to 0.63. It means that if the ratio of the number of malignant patches to the total number of patches is greater than 0.63, this EBUS image is classified as malignant; otherwise, it is classified as benign. The confusion matrix of the multi-patch-based model is depicted in Figure 13c, which indicates that three malignant lesions were misclassified as benign, while two benign lesions were misclassified as malignant.

Figure 16 depicts the misclassification results of the multi-patch-based model. Figure 16a depicts a malignant lesion that was misclassified as benign, while Figure 16b depicts a benign lesion that was misclassified as malignant. The main reason for misclassification of Figure 16a is because the lesion region is too small, which allows only a few patches to be used for classification; whereas the misclassification of Figure 16b is because the texture of the lesion is heterogeneous.

From the statistical results in Table 6, it can be seen that the radiomics feature and patient data-based model and the multi-patch-based model yield high accuracy regardless of the number of image data; whereas the single image-based model yields the lowest accuracy. The main problem of the single image-based model is that CNN needs a very large dataset to obtain good results. However, the texture of the boundary and the surrounding areas of a lesion are also important features, thus this paper proposes to integrate the statistical results of the three models to perform the final classification using the weighted ensemble classification to assign the weight to each model based on the classification performance. In this paper, $w_{1},w_{2}$, and $w_{3}$ were set to 0.41, 0.08, and 0.51, based on the performance of the models. The optimal cutoff was set to 0.53, which is defined by the value that yields the highest accuracy during the training. The lesions were divided into two classes: benign when P(I)≤0.53 and malignant when  P(I)>0.53. The proposed framework’s confusion matrix is depicted in Figure 13d, which indicates that all malignant lesions were correctly classified, while only two benign lesions were misclassified as malignant.

Figure 17 depicts the effectiveness of the proposed framework. Figure 17a shows the correct classification result of the proposed framework while two out of three classification results from three models are incorrect; i.e., the radiomics feature and patient data-based model and the single image-based model misclassified benign lesion as malignant. Figure 17b shows the correct classification result of the proposed framework, while two out of three classification results from three models are incorrect; i.e., the single image-based model and the multi-patch-based model misclassified the malignant lesion as benign.

Table 6 displays the classification performance of all classification models. The proposed framework yields accuracy, sensitivity, specificity, positive predictive value, and a negative predictive value of 95.00, 100, 86.67, 92.59, and 100, respectively. Furthermore, by comparing the ROC curves and AUC values of all classification models in Figure 18, the AUC value obtained by using the proposed framework is 0.9333, which is higher than those of the other three models.

The proposed framework performs well, but it still has some limitations. First, there is no evidence that the proposed framework works well with other types of medical images. Second, the number of patches depends on the region of lesion in each image, thus a lesion with only a few patches can be easily misclassified in a multi-patch-based model.

## 5. Conclusions

In this paper, a novel pulmonary lesion classification framework for EBUS images was proposed by integrating three classification models with the weighted ensemble classification. The proposed framework works well with imbalanced data and small datasets. The radiomics feature and patient data-based model is suitable for any size of the dataset because it classifies a lesion based on both radiomics features and patient data that contain substantial amounts of relevant information, such as texture, shape, size, age, and gender. It also works well for an imbalanced dataset. The single image-based model uses the global characteristics of a lesion from the entire EBUS images to train the model. Thus, the model can learn and extract the dominant features from an image by the model itself. However, the disadvantage of this model is that it needs a large volume of data to obtain good results. On the other hand, the multi-patch-based model uses local characteristics of a lesion from each patch. By integrating these three models with the weighted ensemble classification, the proposed framework can improve the classification results by using both local and global characteristics of a lesion. The proposed framework achieves promising pulmonary lesion classification results and outperforms individual models. Due to ethics concerns, data insufficiency is a common problem in medical applications, and the proposed framework can tackle this problem. In the future, the proposed framework will be tested on different sets of medical images.

## Figures and Tables

**Figure 1 diagnostics-12-01552-f001:**
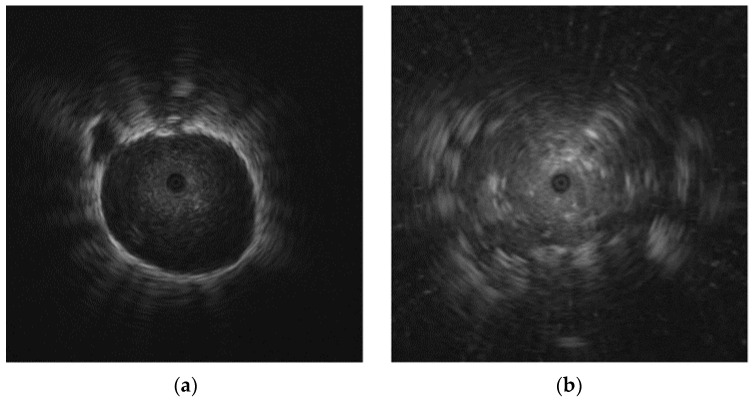
Examples of different pulmonary lesion patterns in EBUS images: (**a**) benign lesion; (**b**) malignant lesion (benign and malignant lesions were confirmed after core needle biopsy).

**Figure 2 diagnostics-12-01552-f002:**
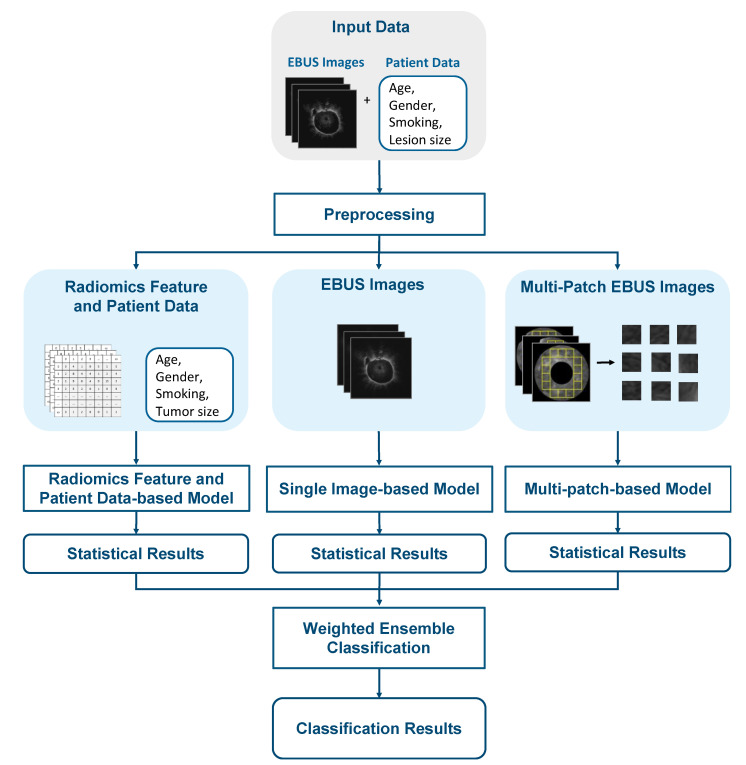
The framework of the proposed pulmonary lesion classification system.

**Figure 3 diagnostics-12-01552-f003:**
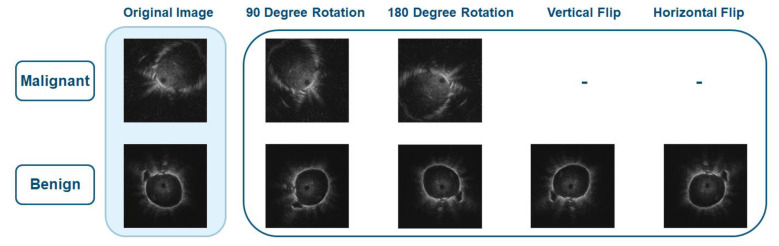
Data augmentation for malignant and benign images.

**Figure 4 diagnostics-12-01552-f004:**
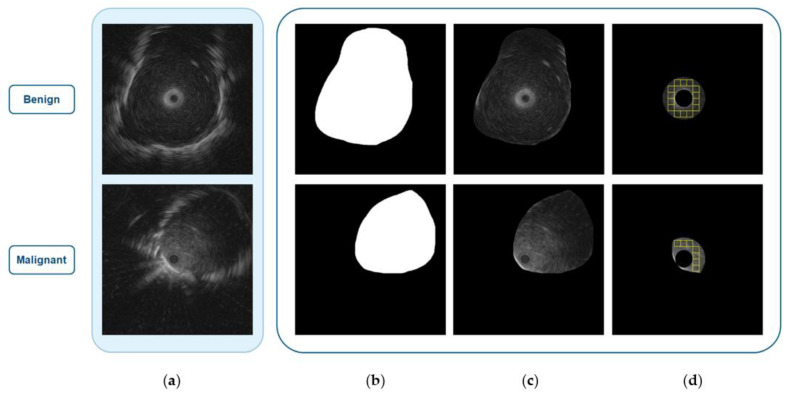
Results from the preprocessing step: (**a**) original images; (**b**) mask images; (**c**) regions of interest; (**d**) WOIs in the intersection area of the region of interest and the ring.

**Figure 5 diagnostics-12-01552-f005:**
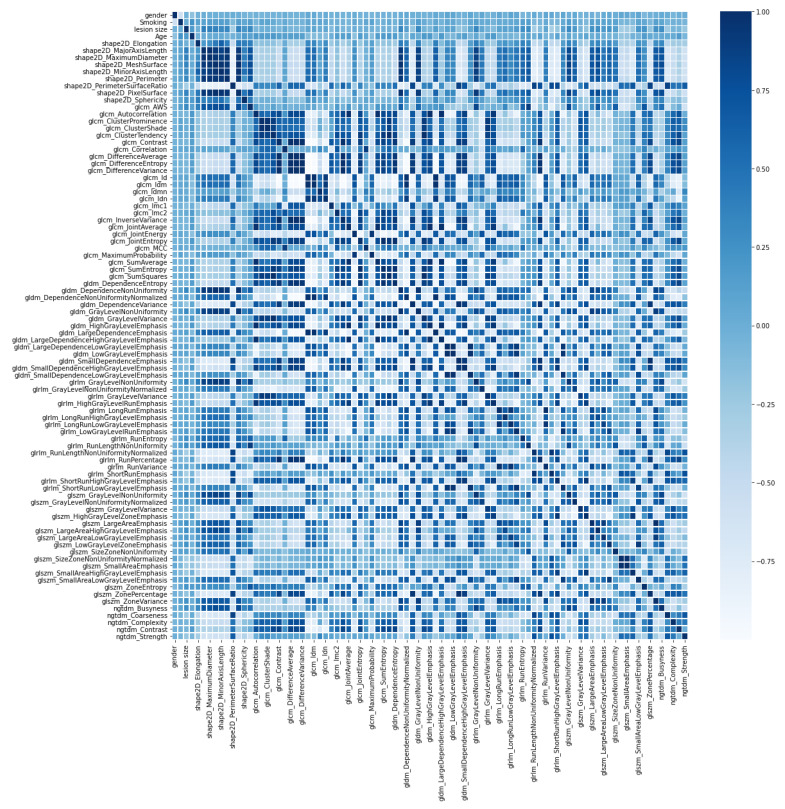
Heat map of Pearson correlation coefficient matrix for all features.

**Figure 6 diagnostics-12-01552-f006:**
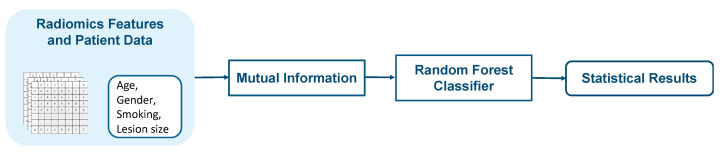
Block diagram of the radiomics feature and patient data-based model.

**Figure 7 diagnostics-12-01552-f007:**
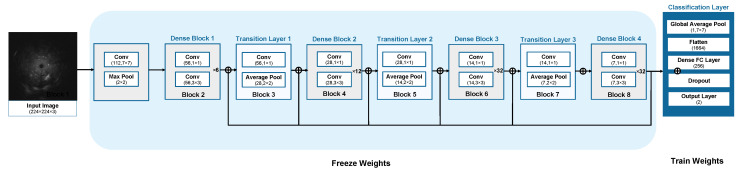
The architecture of the fine-tuned DenseNet 169 for the single image-based model.

**Figure 8 diagnostics-12-01552-f008:**
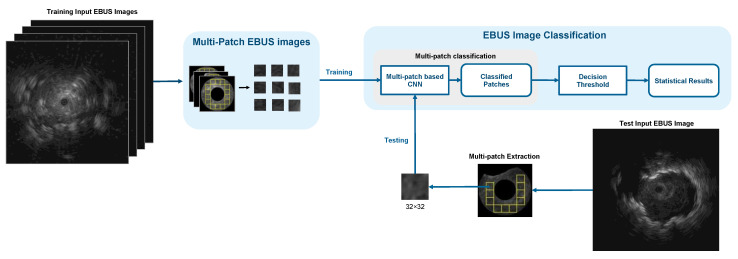
The architecture of the multi-patch-based model.

**Figure 9 diagnostics-12-01552-f009:**
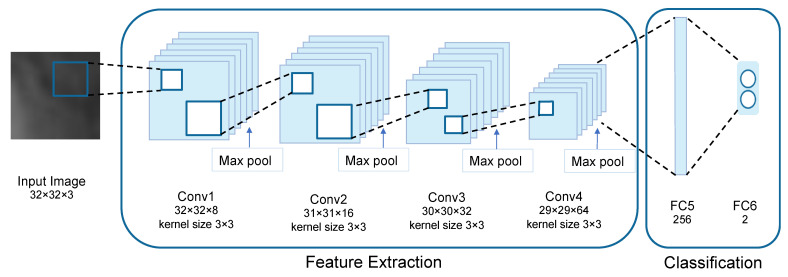
The architecture of the proposed CNN in the multi-patch-based model.

**Figure 10 diagnostics-12-01552-f010:**
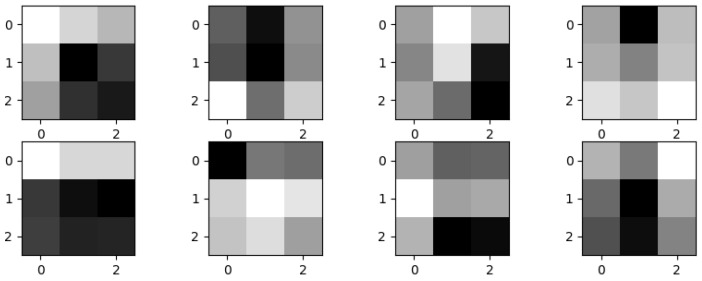
Weights learned by the first convolutional layer.

**Figure 11 diagnostics-12-01552-f011:**
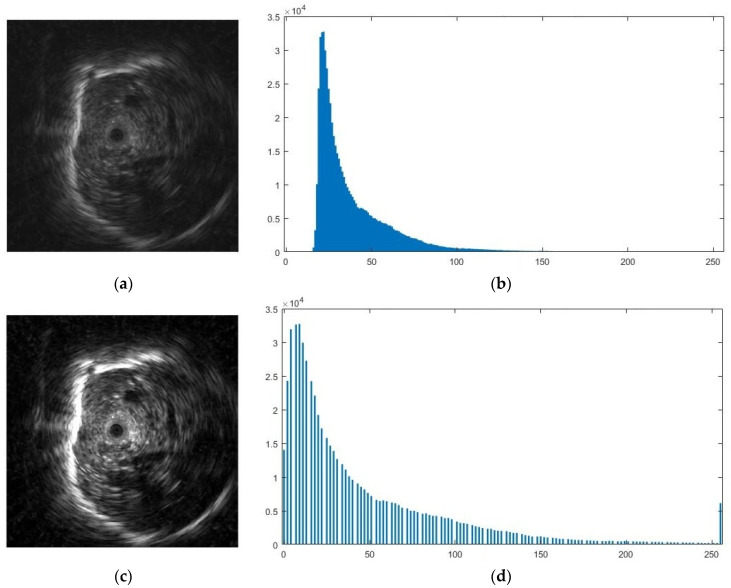
(**a**) The original EBUS image; (**b**) the histogram of (**a**); (**c**) the enhanced EBUS image; (**d**) the histogram of (**c**).

**Figure 12 diagnostics-12-01552-f012:**
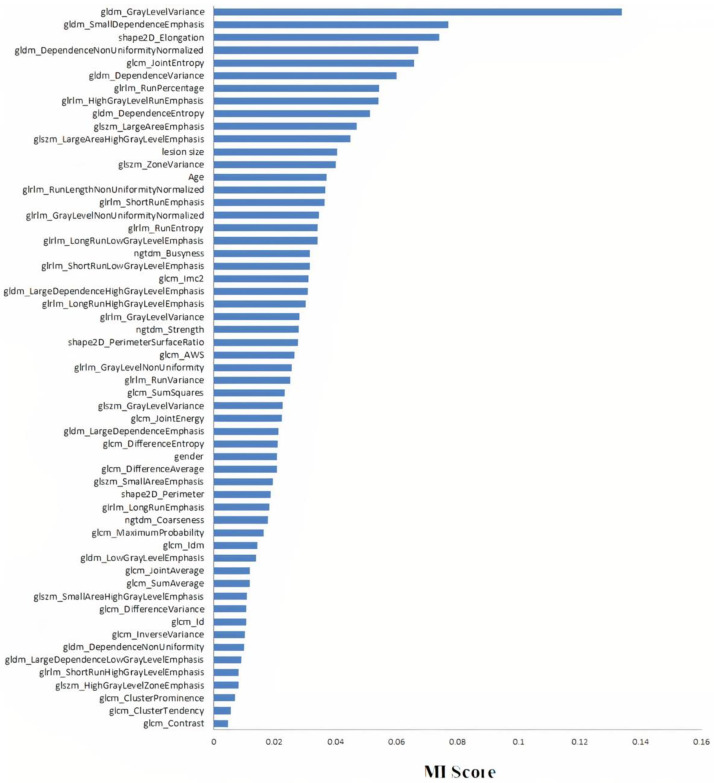
Feature selection of radiomics feature and patient data-based model.

**Figure 13 diagnostics-12-01552-f013:**
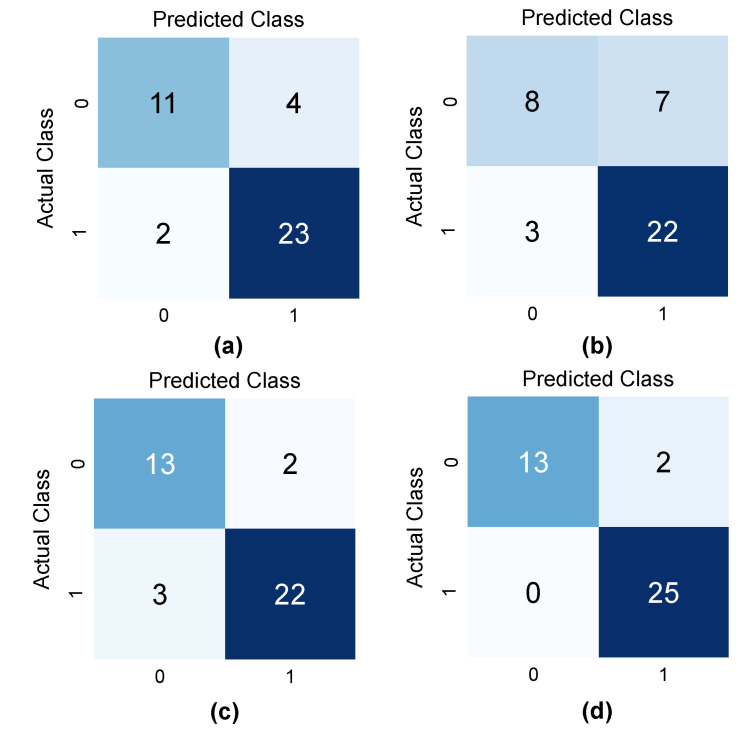
Confusion matrices: (**a**) the radiomics feature and patient data-based model; (**b**) the single image-based model; (**c**) the multi-patch-based model; and (**d**) the proposed framework.

**Figure 14 diagnostics-12-01552-f014:**
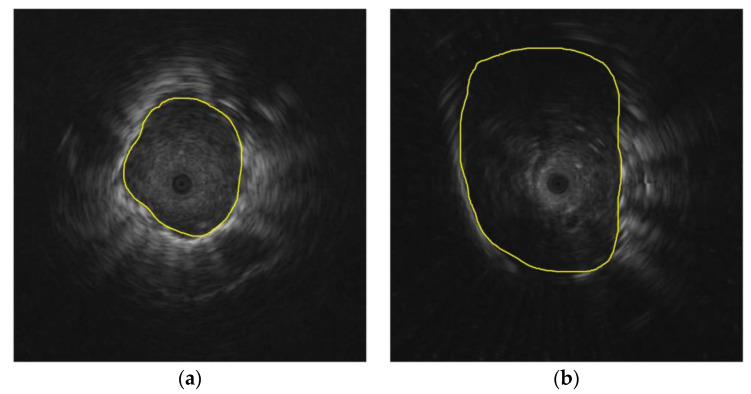
Misclassification of the radiomics feature and patient data-based model: (**a**) malignant lesion was misclassified as benign; (**b**) benign lesion was misclassified as malignant.

**Figure 15 diagnostics-12-01552-f015:**
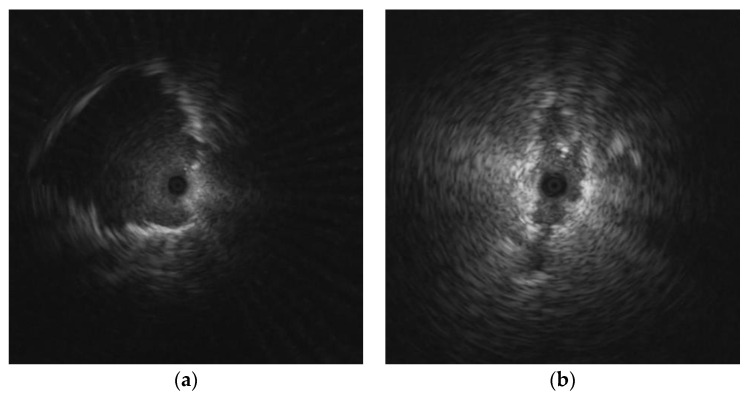
Misclassification of the single image-based model: (**a**) malignant lesion was misclassified as benign; (**b**) benign lesion was misclassified as malignant.

**Figure 16 diagnostics-12-01552-f016:**
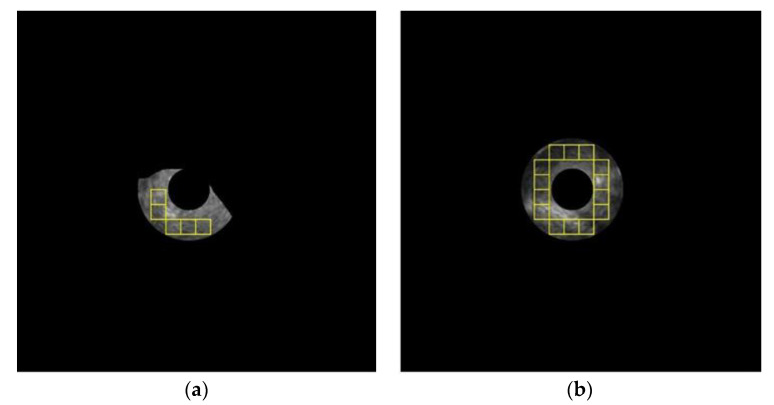
Misclassification of the multi-patch-based model: (**a**) malignant lesion was misclassified as benign; (**b**) benign lesion was misclassified as malignant.

**Figure 17 diagnostics-12-01552-f017:**
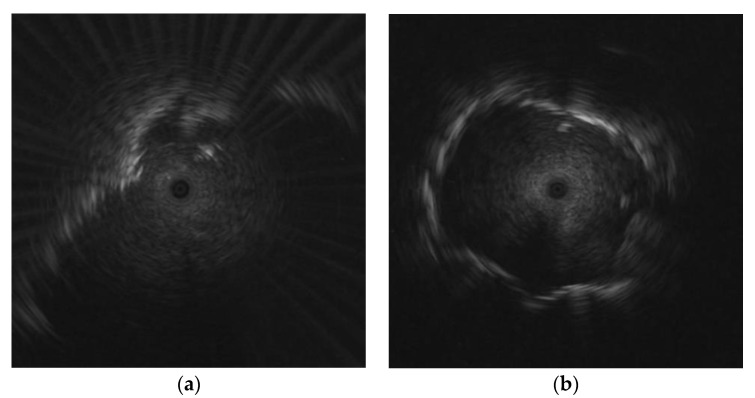
Two out of three models misclassified a lesion, but the proposed framework can classify correctly: (**a**) benign lesion was misclassified as malignant in radiomics feature and patient data-based model and multi-patch-based model; (**b**) benign lesion was misclassified as malignant in single image-based model and multi-patch-based model.

**Figure 18 diagnostics-12-01552-f018:**
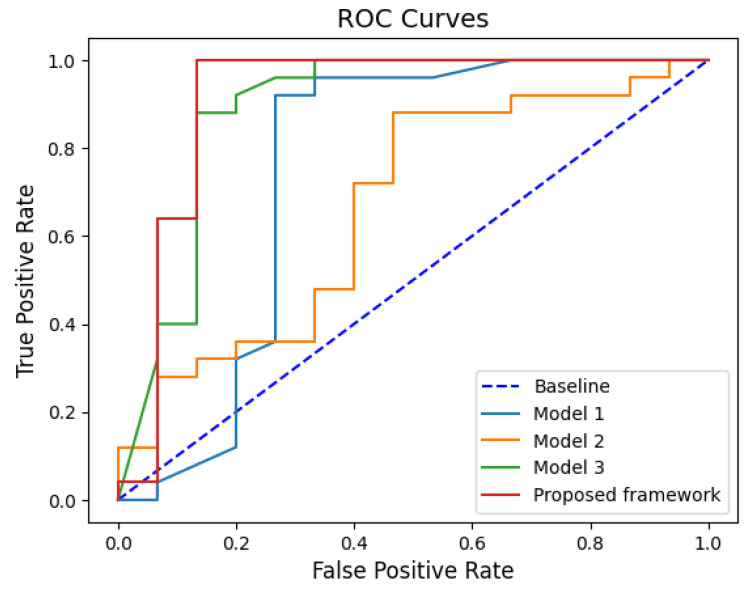
ROC curves and AUC values of the classification models.

**Table 1 diagnostics-12-01552-t001:** Clinical details of the patients.

	Malignant	Benign
Number of patients	124 (74 male, 50 female)	76 (29 male, 47 female)
Age (Mean ± SD)	64.32 ± 13.21	57.63 ± 15.51
Lesion size	≥3 cm (75), <3 cm (49)	≥3 cm (38), <3 cm (38)
Smoking History	non-smoking (52), smoking (35), ex-smoking (37)	non-smoking (29), smoking (27), ex-smoking (20)

**Table 2 diagnostics-12-01552-t002:** Distribution of EBUS images after class balancing.

	Malignant	Benign	All
Original EBUS image data	99	61	160
Augmented image data	198	244	442
Total of training image data	297	305	602

**Table 3 diagnostics-12-01552-t003:** The hyper-parameters of DenseNet 169 architecture.

Hyper-Parameter	Value
Optimizer	Stochastic Gradient Descent
Learning rate	0.0001
Loss function	Cross-entropy
Batch size	32

**Table 4 diagnostics-12-01552-t004:** The configuration of CNN for the multi-patch-based model.

Layer	Type	Kernel Size	Stride	Output Size
Data	Input			3 × 32 × 32
Conv1	Convolution	3 × 3	1	8 × 32 × 32
Conv2	Convolution	3 × 3	1	16 × 31 × 31
Conv3	Convolution	3 × 3	1	32 × 30 × 30
Conv4	Convolution	3 × 3	1	64 × 29 × 29
FC5	Fully connected			256 × 1 × 1
FC6	Fully connected			2 × 1 × 1

Every convolutional layer is followed by pooling layer. The ReLU activation function is not shown for brevity.

**Table 5 diagnostics-12-01552-t005:** The hyper-parameters of CNN architecture for the multi-patch-based model.

Hyper-Parameter	Value
Optimizer	Adam
Learning rate	0.001
Loss function	Cross-entropy
Batch size	128

**Table 6 diagnostics-12-01552-t006:** The classification performance of different classification models.

	Acc (%)	Sen (%)	Spec (%)	PPV (%)	NPV (%)	AUC
Radiomics feature and patient data-based model	85.00	92.00	73.33	85.19	84.62	0.8267
Single image-based model	75.00	88.00	53.33	75.86	72.72	0.7067
Multi-patch-based model	87.50	88.00	86.67	91.67	81.25	0.8733
Proposed framework	**95.00**	**100**	**86.67**	**92.59**	**100**	**0.9333**

The values in bold font indicate the best index values.

## Data Availability

Not applicable.

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
