# Peer review of "Pulmonary Lesion Classification Framework Using the Weighted Ensemble Classification with Random Forest and CNN Models for EBUS Images"

_diagnostics, 2022, doi:10.3390/diagnostics12071552_

Round 1

Reviewer 1 Report

In this work, the authors presented an ensemble classification model by combing the random forest and deep learning methods for pulmonary lesion classification on EBUS Images. The authors specifically designed the model for small-size data, which is an important and challenging issue in computer-assisted diagnosis. The results on 200 cases show the promising results of the proposed framework. I think this paper could be further improved if the following concerns can be addressed.

1. The proposed CNN model did not contain any pooling layers. Could the authors comment on this? Pooling layers could reduce the size of feature maps and prevent overfitting.

2. After the training of CNN, interpretability is an essential step to understanding how the decision was made. I am wondering if the authors could provide some heatmaps for visualizations.

3. The ROC in Figure 17 looks weird to me. It is either curve or multi-steps rather than the way it shows now.

4. Some references using ensemble learning or towards interpretability for lung cancers should be discussed. Some examples are

a) "An ensemble learning method based on ordinal regression for COVID-19 diagnosis from chest CT"

b) "Strided Self-Supervised Low-Dose CT Denoising for Lung Nodule Classification"

c) "Shape and margin-aware lung nodule classification in low-dose CT images via soft activation mapping"

Author Response

Response to Reviewer 1 Comments

First, we would like to thank you for your valuable comments. Your comments can help us improve the manuscript. We respond to your comments below:

In the manuscript, we use blue highlight for reviewer 1 and yellow highlight for reviewer 2.

Point 1: The proposed CNN model did not contain any pooling layers. Could the authors comment on this? Pooling layers could reduce the size of feature maps and prevent overfitting.

Response 1:

The proposed CNN model contains Max pooling layers. The detail of Max pooling layer was added in line 335 and figure 9.

Point 2: After the training of CNN, interpretability is an essential step to understanding how the decision was made. I am wondering if the authors could provide some heatmaps for visualizations.

Response 2: heatmaps for visualization were added in Figure 10 followed by the detail.

Point 3: The ROC in Figure 17 looks weird to me. It is either curve or multi-steps rather than the way it shows now.

Response 3: We have modified Figure  17 according to your suggestions.

Point 4: Some references using ensemble learning or towards interpretability for lung cancers should be discussed. Some examples are

  1. a) "An ensemble learning method based on ordinal regression for COVID-19 diagnosis from chest CT"

  1. b) "Strided Self-Supervised Low-Dose CT Denoising for Lung Nodule Classification"

  1. c) "Shape and margin-aware lung nodule classification in low-dose CT images via soft activation mapping"

Response 4:  We have included the discussion on using ensemble learning according to your suggestions in lines 134 to line 153. 

Reviewer 2 Report

I have some minor issues:

1- Line 38: early on !!! not completed sentence.

2- Data from 2011 to 2016, You did not find newer, did you?

3- Why you just chose 80:20 training:testing ratio, why not 70:30 or 60:40 Justify?

4- In equation 2: what is (??') why did not you write ??/ ? ?

5- In equation 7: is it total benign or total malignant?

6- Paragraph in lines 451-453 (From the statistical results in Table 10, it can be seen that single image-based model 451 and multi-patch-based model yield high accuracy regardless of the number of image data; 452 whereas single image-based model yields the lowest accuracy.) needs revision.

7- How you set the optimal cut off to 0.53?

8- Conclusion should not include new information like what you write in this sentence : (These local characteristics focus on the patterns of texture; i.e., homogeneity, heterogeneity, hyperechoic dot, hyperechoic arc, anechoic area, and linear air bronchogram.)

Author Response

Response to Reviewer 2 Comments

First, we would like to thank you for your valuable comments. Your comments can help us improve the manuscript. We respond to your comments below:

In the manuscript, we use blue highlight for reviewer 1 and yellow highlight for reviewer 2.

Point 1: Line 38: early on !!! not completed sentence.

Response 1: we have already corrected it in lines 40-41.

Point 2: Data from 2011 to 2016, You did not find newer, did you?

Response 2: The dataset was obtained from a doctor whom I had a research collaboration with. She studied a specialized program in pulmonary lesions from 2011 to 2016 and after she graduated, I have no one to verify new datasets for me.

Point 3: Why you just chose 80:20 training:testing ratio, why not 70:30 or 60:40 Justify?

Response 3:

Empirical studies show that the best results are obtained if we use 20-30% of the data for testing, and the remaining 70-80% of the data for training.

A commonly used ratio is 80:20, which means 80% of the data is for training and 20% for testing. Other ratios such as 70:30, 60:40, and even 50:50 are also used in practice.

There does not seem to be a clear guidance on what ratio is best or optimal for a given dataset. The 80:20 split draws its justification from the well-known Pareto principle, but that is again just a thumb rule used by practitioners.

We had also conducted the experiments using  70:30 and 60:40 as well, and the best results can be obtained by using 80:20.

Point 4: In equation 2: what is (??') why did not you write ??/ ? ?

Response 4: The symbol “    ‘   ”  is meant to be a comma, thus I have removed it in order to avoid any confusion. 

 Point 5: In equation 7: is it total benign or total malignant?

Response 5: we have already corrected it in equation 7.

Point 6: Paragraph in lines 451-453 (From the statistical results in Table 10, it can be seen that single image-based model 451 and multi-patch-based model yield high accuracy regardless of the number of image data; 452 whereas single image-based model yields the lowest accuracy.) needs revision.

Response 6: we have already corrected it in lines 485-488.

Point 7: How you set the optimal cut off to 0.53?

Response 7:  The optimal cutoff value is defined by the value that yields the highest accuracy during the training in lines 494-495.

Point 8: Conclusion should not include new information like what you write in this sentence : (These local characteristics focus on the patterns of texture; i.e., homogeneity, heterogeneity, hyperechoic dot, hyperechoic arc, anechoic area, and linear air bronchogram.)

Response 8: we have already corrected it by adding more detail in lines 294-298 and 322-324.
